# Treatment-induced increase in total body potassium in patients at high risk of ventricular arrhythmias; a randomized POTCAST substudy

Ulrik Winsløw[1]*, Tharsika Sakthivel[1], Chaoqun Zheng[1], Helle Bosselmann[4], Ketil Haugan[4], Niels Bruun[4,5], Charlotte Larroudé[3], Kasper Iversen[3,5], Hillah Saffi[1], Emil Frandsen[1], Peter Oturai[2], Holger Jan Jensen[2], Michael Vinther[1], Niels Risum[1], Henning Bundgaard[1,5], Christian Jøns[1]

1 Department of Cardiology, Copenhagen University Hospital–Rigshospitalet, Copenhagen, Denmark, 2 Department of Clinical Physiology and Nuclear Medicine, Copenhagen University Hospital–Rigshospitalet, Copenhagen, Denmark, 3 Department of Cardiology, Copenhagen University Hospital, Herlev, Gentofte, Denmark, 4 Department of Cardiology, Zealand University Hospital, Roskilde, Denmark, 5 Department of Clinical Medicine, University of Copenhagen, Copenhagen, Denmark

* ulrik.christian.winsloew.01@regionh.dk

**Data Availability Statement:** Data cannot be shared publicly because of restrictions by Danish legislation (General Data Protection Regulation –

## Abstract

### Objective

Hypokalemia is associated with increased risk of arrhythmias and it is recommended to monitor plasma potassium (p-K) regularly in at-risk patients with cardiovascular diseases. It is poorly understood if administration of potassium supplements and mineralocorticoid receptor antagonists (MRA) aimed at increasing p-K also increases intracellular potassium.

### Methods

Adults aged≥18 years with an implantable cardioverter defibrillator (ICD) were randomized (1:1) to a control group or to an intervention that included guidance on potassium rich diets, potassium supplements, and MRA to increase p-K to target levels of 4.5–5.0 mmol/l for six months. Total-body-potassium (TBK) was measured by a Whole-Body-Counter along with p-K at baseline, after six weeks, and after six months.

### Results

Fourteen patients (mean age: 59 years (standard deviation 14), 79% men) were included. Mean p-K was 3.8 mmol/l (0.2), and mean TBK was 1.50 g/kg (0.20) at baseline. After six-weeks, p-K had increased by 0.47 mmol/l (95%CI:0.14;0.81), p = 0.008 in the intervention group compared to controls, whereas no significant difference was found in TBK (44 mg/kg (-20;108), p = 0.17). After six-months, no significant difference was found in p-K as compared to baseline (0.16 mmol/l (-0.18;0.51), p = 0.36), but a significant increase in TBK of 82 mg/kg (16;148), p = 0.017 was found in the intervention group compared to controls.

GDPR). Data are available from the a non-author data access committee for researchers who meet the criteria for access to confidential data. Requests can be send to: Anna Kirstine Ringgaard, MSc, Ph.d Email: anna.kirstine.ringgaard@regionh. dk Doctor Ringgaard is manager/head of a data access committee at Rigshospitalet, Denmark. Her titles include project administrator and research coordinator which she is for several projects at Rigshospitalet and handles legal affairs as well as funding.

**Funding:** This study was supported by: The Danish Council for Independent Research Grant recipient: CJ Grant Number: 8020-00399B Url: https://dff.dk/ en The Hartmann Foundation Grant recipient: HB Grant Number: 2019 Url: https://www. hartmannfonden.dk/english/ The Danish Heart Foundation Grant recipient: HB Grant Number: 2019 Url: https://hjerteforeningen.dk/ Snedkermester Sophus Jacobsen og hustru Astrid Jacobsens Fond Grant recipient: HB Grant Number: 2019 Url: https://sophusjacobsenfond.dk/ The Novo Nordisk Foundation Grant recipient: NR Grant Number: NNF20OC0064048 Url: https:// novonordiskfonden.dk/ The funders had no role in study design, data collection and analysis, decision to publish, or preparation of the manuscript.

**Competing interests:** The authors have declared that no competing interests exist.

## Conclusions

Increased potassium intake and MRAs increased TBK gradually and a significant increase was seen after six months. The differentially regulated p-K and TBK challenges current knowledge on potassium homeostasis and the time required before the full potential of p-K increasing treatment can be anticipated.

## Trial registration

www.clinicaltrials.gov (**NCT03833089**).

## Introduction

Plasma potassium (p-K) is one of the most commonly measured parameters in the healthcare sector [1]. Detrimental effects of changes in p-K are seen in numerous conditions and disorders and may be induced by several drugs [2–4]. Notably, hypokalemia has been reported in up to 20% of hospitalized patients and is associated with significantly increased risk of developing malignant ventricular arrythmias and atrial fibrillation [5, 6]. Patients with certain cardiovascular diseases are particularly sensitive to hypokalemia [2, 7]. The general importance of the potassium homeostasis is underscored by observational studies, drug trials as well as a recent randomized trial showing that increased potassium intake improves survival [8–10], whereas drugs that induce potassium loss such as non-potassium sparing diuretics have been shown to increase the risk of both supraventricular and ventricular arrhythmias [11, 12]. Overall, a U shaped relationship has been found between p-K and mortality in several observational studies of patients with heart failure as well as in the general population that consistently show lowest risk around the mid-normal to high-normal p-K levels [13–15].

In myocardial cells, changes in extracellular and intracellular potassium levels modulates cellular membrane potential, depolarization velocity and automaticity, as well as repolarization time and refractory periods, in particular in patients with reduced repolarization reserve [2]. Approximately 98% of total body potassium is located intracellularly and the transmembraneous concentration gradient is regulated by ion channels and the Na,K ATPase. The activity of the Na,K-ATPase is regulated by feedback-loops that control fluid balance, adrenergic and glycemic levels as well as pH buffer systems, that are able to rapidly shift potassium between the extracellular and the intracellular space [16–18]. These minute-to-minute regulations make p-K a volatile parameter and likely a poor indicator of total body potassium (TBK). TBK includes potassium in bones and other structural tissues with slower metabolism, and if TBK deposits increase at a slow rate, it may be necessary to continue potassium-increasing therapy for an extended period to reach new steady state levels. In contrast, if TBK is not increased, the clinical effect of potassium supplementation may mainly be related to the changes in p-K and should be adjusted accordingly. Improved knowledge of the regulation of TBK may disclose novel targets for interventions in disorders and conditions known to be sensitive to changes in p-K.

A Whole-Body Counter (WBC) approach is used to accurately estimate human TBK *in vivo*. The current study was designed to determine if TBK can be increased actively using dietary guidance, potassium supplements and mineralocorticoid receptor antagonists (MRA) and to determine how the rate and magnitude of these changes correlate to changes in p-K.

## Materials and methods

### Population

This single-center study was conducted between January and December 2021 as a substudy to the POTCAST trial at the Copenhagen University Hospital, Rigshospitalet, Copenhagen, Denmark. Details, background and design of the ongoing POTCAST trial have been published previously [19]. The purpose of the POTCAST trial is to investigate if high-normal potassium level reduces the incidence of malignant tachyarrhythmias and death. In brief, the POTCAST trial is a randomized, open-labelled, clinical trial enrolling 1,000 patients at high-risk of malignant arrhythmias as defined by clinically driven treatment with a primary or secondary preventive implantable cardioverter defibrillator (ICD). Patients are randomized (1:1) through a concealed computer-generated sequence (project-RedCap.org), to either the control group or an intervention that includes a potassium-rich dietary guidance and daily intake of oral potassium supplements and MRAs with the aim to increase and maintain p-K at target levels of 4.5–5.0 mmol/l. In the current open-labelled substudy consecutive participants from the POTCAST trial were included.

Inclusion criteria for the main trial are a baseline p-K $\leq$4.3 mmol/l, age $\geq$18 years, and treatment with an ICD in accordance with clinical guidelines. Patients are excluded if they have severe renal failure (estimated glomerular filtration rate below 30 ml/min/1.73 $m^2$), are pregnant, or are unable to provide informed consent.

Criteria for inclusion in the present substudy were similar, except a baseline p-K $\leq$4.0 mmol/l. We excluded patients with claustrophobia and patients undergoing radiation treatment (e.g., radioiodine therapy) six months prior or nuclear medicine imaging three months prior to inclusion to avoid interference with the Whole-Body-Counter.

### Screening for inclusion and patient flow

Patients included in the main study were screened with baseline blood sampling (hemoglobin, p-$K^+$, p-$Na^+$, p-$Ca^{2+}$, p-$Mg^{2+}$, creatine, and aspartate transaminase), blood pressure measurement, ECG recordings, and echocardiography. Patients fulfilling in- and exclusion criteria in the present substudy were scheduled for baseline measurements of TBK by Whole-Body-Counter after written informed consent was obtained. Baseline measurements of TBK were done twice for each patient on two consecutive days as soon as possible after screening to reduce noise-variance. Patients were then randomized 1:1 to either the intervention- or control group after the first set of TBK measurements had been obtained to ensure that baseline measurements were performed before any intervention or group-specific behavioral changes. The intervention was then initiated and follow-up TBK measurements were performed six weeks and six months after baseline measurement in both the control and intervention group along with repeated blood laboratory testing. Whenever possible, the follow-up visits were scheduled at approximately the same time of day as the baseline visit for each patient, to reduce the potential effects of circadian variation or the pharmacokinetics of the other daily medications on TBK and p-K measurements. All TBK and p-K analyses were done by technicians blinded to the patient's study group assignment. A CONSORT flow-chart of patient inclusion and follow-up is shown in Fig 1.

### Intervention

Between baseline and the six-week follow-up, the intervention group was given guidance on intake of a potassium rich diets along with potassium supplements and MRA (spironolactone or eplerenone) with the aim to increase and maintain p-K at target levels between 4.5–5.0

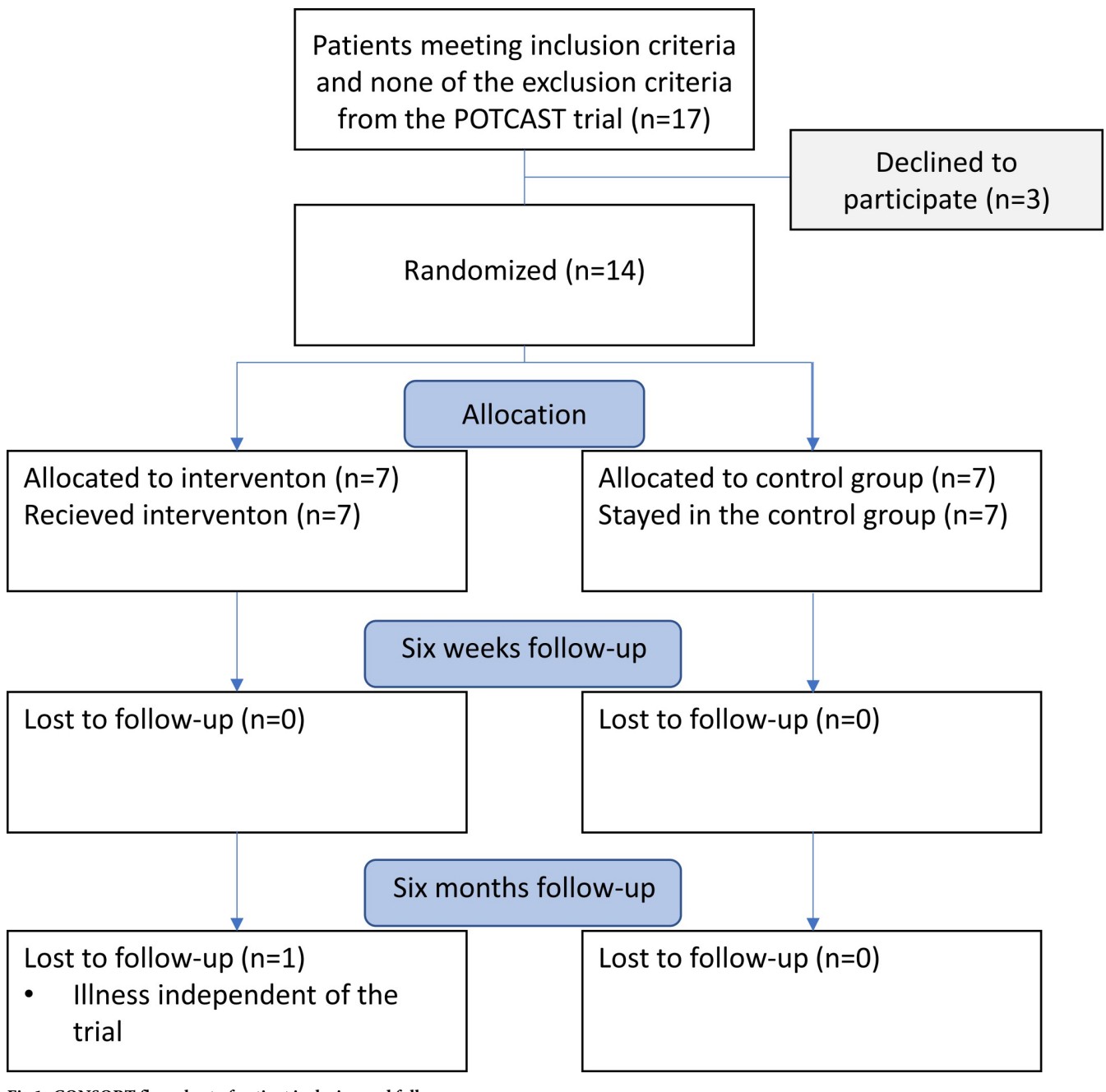

**Fig 1. CONSORT flow-chart of patient inclusion and follow-up.**

mmol/l according to the design of the POTCAST trial [19]. Patients were uptitrated until target p-K was reached or until maximally tolerated doses of potassium supplement (up to 4,500 mg ~ 60 mmol daily) and MRA (up to 100 mg daily) were given. Blood pressure and renal function were monitored during up-titration. Patients were asked to maintain the reached dose throughout the duration of the trial. Patients randomized to the control group continued usual standard of care. Compliance was assessed by questioning the patients about drug intake at each visit and this was confirmed by prescription fillings through electronic medical records.

## Main outcome measures

Between- group difference in changes in TBK and p-K from baseline to six weeks, and from baseline to six months.

## Whole-Body-Counter $^{40}$K measurement

The Whole-Body Counter (WBC) procedure exploits that naturally occurring potassium exists as three different isotopes; $^{39}$K, $^{40}$K, and $^{41}$K with exact mass percentages of 93.08%, 0.0118% and 6.91%, respectively [20]. The radioactive isotope $^{40}$K emits gamma-rays which can be detected by a WBC. The mass percentage distribution of 0.0118% of $^{40}$K allows for accurate extrapolation to TBK from these WBC measurements.

In the present study, TBK was estimated using the whole-body scintillation counter (Nuclear Enterprises Ltd, Sighthill, Edinburgh, Scotland) [21, 22] at the Copenhagen University Hospital, Rigshospitalet. During the scan, four Thallium doped Sodium Iodide scintillation detectors placed in a lead-lined steel chamber were placed over the subject's chest, over the abdomen, below the neck and below the thighs. The WBC was calibrated with phantoms consisting of plastic containers filled with known precise quantities of potassium chloride solution (5 g/L). These phantoms were individually adjusted in weight and height to the patients being measured [23, 24]. Background radiation was measured before and after each patient measurement.

The patients showered and changed to hospital gowns before entering the lead- and steel-shielded chamber to avoid interference from external contamination. In the chamber the patient was placed in a supine position for 30 minutes while being measured by the WBC. $^{40}$K decays directly to $^{40}$Ar with the emission of 1.46 million electron volt (MeV) gamma-rays which is detected by the WBC. The examination resulted in an energy spectrum showing number of registered gamma-rays (S1 Fig in S1 File). The amount of $^{40}$K in a patient was calculated from AUC analyses (ABACOS-2000, v1.3E, Canberra) at 1.46 MeV after subtraction of background counts. TBK was calculated assuming that $^{40}$K constitutes 0.0118% of the naturally occurring potassium in the biosphere.

## Ethics

The study was approved by the Regional Danish Committee on Health Research Ethics (Regional Videnskabsetisk Komité)—approval no. H-18044908 and the Danish Data Protection Agency—approval no. VD-2018-453. The study was conducted in accordance with the declaration of Helsinki. The limited size of the study was solely based on the sample size calculation as it is, due to ethical considerations, recommended to avoid exposing more patients to trial examinations than what has been shown to be necessary to reach a conclusion.

## Statistics

Values are summarized as mean and standard deviation or as median and interquartile range as appropriate. Students t-test or the Kruskal Wallis test were used to compare continuous data. Data was fitted using mixed-effect linear regression models with random intercepts and patient ID as random effect to test the primary hypothesis. A Bland-Altman plot was used to interpret comparability of the two baseline TBK measurements. The assumption of normal distribution of model residuals was tested by Shapiro-Wilk test. For changes in p-K the W statistic was 0.97, P = 0.39 and for TBK the W statistic was 0.96, P = 0.11. Analyses were carried out using standard statistical software (R version 4.1.0).

## Sample size calculations

No other observational or clinical trials have investigated the effect sizes of the combination of changes in dietary potassium intake with potassium supplementation and MRA on TBK. However, based on an earlier trial on potassium supplements the study was powered to detect an increase in TBK of 5% [23]. An increase in TBK of 5% from an average of 1.6 g potassium/kg body weight with a standard deviation of the paired difference of 0.05 g potassium/kg body weight a total of 12 patients (6 in each group) were required (two-tailed test, 1-β = 0.8, α = 0.05). To compensate for potential loss-to-follow-up a total of 14 patients were included.

# Results

## Baseline

Seven patients were randomized to each group. Baseline characteristics of the participants included in the two groups are summarized in Table 1. The mean age was 59 years (standard deviation [SD] 14), 11 out of 14 were male, and mean body weight was 89 kg (SD 13). The intervention- and control group were fairly well matched. Of note, the patients had a wide variety of ICD indications. Only one patient in each group was treated with non-potassium sparing diuretics at baseline.

From baseline to six-weeks follow-up a mean dose of 61 mg (SD 28) MRA and a mean dose of potassium supplement of 2,678 mg (SD 1,048) ~ 36 mmol were prescribed to the

**Table 1. Baseline characteristics for patients in the intervention (n = 7) and control group (n = 7).**

|  | Intervention group | Control group |
|---|---|---|
| **Age, years** | 57 (17) | 61 (12) |
| **Male gender, n (%)** | 4 (57) | 7 (100) |
| **Body weight, kg** | 85 (16) | 94 (8) |
| **ICD indication, n (%)** |  |  |
| IHD | 1 (14) | 3 (43) |
| HCM | 1 (14) | 1 (14) |
| DCM | 1 (14) | 0 (0) |
| ARVC | 3 (43) | 0 (0) |
| LQTS | 0 (0) | 1 (14) |
| IVF | 1 (14) | 2 (28) |
| **Medication, n (%)** |  |  |
| Beta-blocker | 7 (100) | 7 (100) |
| ACE inhibitor or ARB | 2 (29) | 4 (57) |
| MRA | 0 (0) | 1 (14) |
| Non-potassium sparing diuretic | 1(14) | 1(14) |
| Potassium supplement | 1 (14) | 1 (14) |
| **P-Potassium, mmol/l** | 3.8 (0.1) | 3.8 (0.2) |
| **P-Sodium, mmol/l** | 140 (2.6) | 141 (1.9) |
| **P-Magnesium, mmol/l** | 0.86 (0.05) | 0.86 (0.10) |
| **P-Creatinine, µmol/l** | 73 (15) | 80 (11) |

Summarized by mean (standard deviation) or number (percent).

ARVC: Arrhythmogenic right ventricular cardiomyopathy, DCM: Dilated cardiomyopathy, HCM: Hypertrophic cardiomyopathy, IHD: Ischemic heart disease, IVF: Idiopathic VF

LQTS: Long QT syndrome.

participants in the intervention group. There was an increase in p-K in all patients in the intervention group and three out of seven patients reached target levels of p-K between 4.5–5.0 mmol/l.

One patient was diagnosed with another illness independent of the trial between the six-week- and the six-month follow-up. Due to subsequent diagnostic nuclear isotope imaging that would have affected the WBC measurement, the patient was excluded from the six-month follow-up.

At baseline, mean p-K was 3.8 mmol/l (SD 0.1) in the intervention group and 3.8 mmol/l (SD 0.2) in the control group (P = 0.44). Baseline TBK was 1.44 g/kg (SD 0.23) in the intervention group and 1.56 g/kg (SD 0.17) in controls (P = 0.33).

## Six-week follow-up

At six-weeks follow-up mean p-K was 4.5 mmol/l (SD 0.4) in the intervention group and 3.9 mmol/l (SD 0.4) in the control group, i.e., p-K was significantly increased in the intervention group compared to the control group with a mean difference in changes of 0.47 mmol/l (95% CI: 0.14; 0.81), P = 0.008. After six weeks, mean TBK was not significantly increased in the intervention group compared to the control group, a mean difference in changes of 0.044 g/kg (-0.02; 0.11), P = 0.17.

## Six-month follow-up

At six-months follow-up, TBK was significantly increased in the intervention group compared to the control group with mean difference in changes of 0.082 g/kg (0.016; 0.148), P = 0.017 from baseline (Fig 2A). There were no significant changes in p-K (mean difference 0.16 mmol/l (-0.18; 0.51), P = 0.36 compared to baseline (Fig 2B). The mean TBK and p-K in each study group at baseline, six weeks, and six months are shown in S1 Table in S1 File. There was no difference observed in plasma sodium or plasma magnesium at the six-week or the six-month follow-up compared to baseline (S2 Table in S1 File).

## Reproducibility of TBK measurements with the $^{40}$K-WBC approach

At baseline, TBK was measured on two consecutive days in each patient. The mean difference between these measurements was 0.024 g/kg body weight (-0.20; 0.07), P = 0.26. Intraclass correlation: 0.93, P<0.001. A Bland-Altman plot is shown in S2 Fig in S1 File.

## Discussion

In this study we showed an increase in TBK over a six-month period using dietary guidance, oral potassium supplements and MRAs in patients with normokalemia at baseline. Furthermore, changes in p-K did not reflect changes in TBK in a simple manner as p-K increased significantly over weeks and was attenuated over months of continued treatment, while TBK increased gradually over the six-month study period. The results indicate the time it takes to accumulate intracellular potassium.

The observed increase in TBK was significantly larger than it would be, if only extracellular potassium levels were increased. Thus, in an adult, the total amount of potassium in the extracellular fluid is on average 50–75 mmol ~ 1,950–2,925 mg. The estimated increase in TBK found in the present study of 82 mg/kg corresponds to 158 mmol ~ 6,150 mg of potassium in a person weighing 75 kg, i.e., more than twice the total amount of potassium in the extracellular space. This, in combination with only a small increase in p-K, makes it is reasonable to conclude that the vast amount of the increase in TBK was accumulated intracellularly.

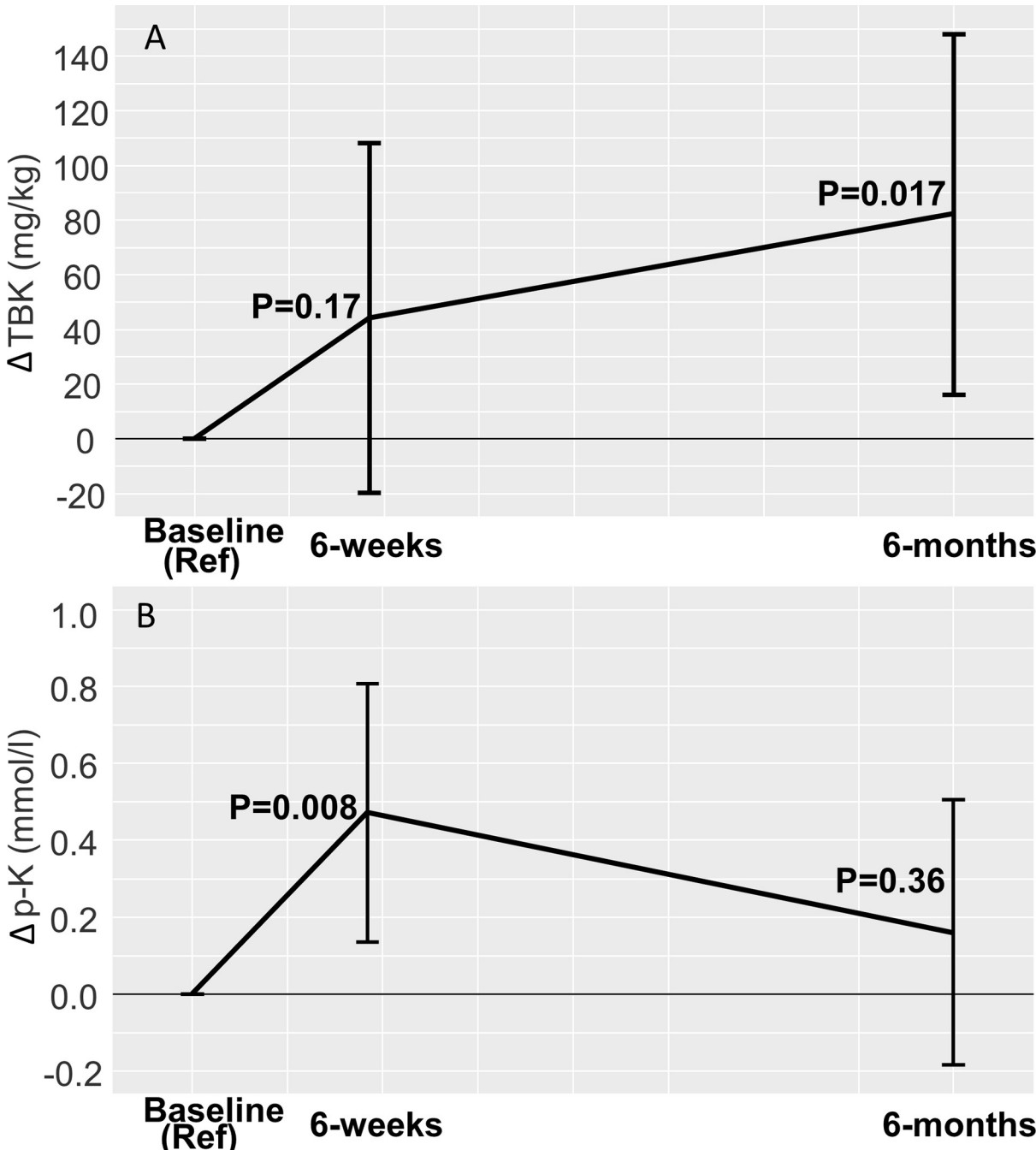

**Fig 2. Difference in changes between groups from baseline (reference) at six-weeks, and six-months with 95% intervals of confidence from a mixed linear random effects regression model.** Panel A: TBK, mg/kg. Panel B: p-K, mmol/l.

No other trial has tested the effects of dietary guidance, potassium supplementation and MRAs on TBK, but a few earlier trials have investigated the effects of potassium supplementation when added to non-potassium sparing diuretics. In 1974, Dargie et al. studied groups of six to eight middle-aged patients and showed a non-significant reduction in TBK from 116.6 g (SD 11.4) to 115.8 g (SD 10.2) after four months when potassium supplements was given in combination with non-potassium sparing diuretics [25]. In 1977, MacLennan et al. showed a

non-significant decrease in TBK from 57.1 mmol potassium/kg lean-body mass (SD 6.3) to 55.8 mmol potassium/kg lean-body mass (SD 5.8) after three months of potassium supplementation in healthy elderly patients (n = 13) [26]. In those studies, however, a relatively small dose of potassium supplement of 24 mmol and 36 mmol daily, respectively was administered. In 1984, Potter et al. showed that in elderly patients (n = 9) with heart failure TBK increased significantly from 1,975 mmol (SD 148) to 2,103 mmol (SD 144) after treatment with 48 mmol potassium supplementation daily when added to treatment with non-potassium sparing diuretics for at least one month [23]. Of notice, no randomized controls were included in any of these studies. These findings indicate that a significant increase in TBK in patients treated with non-potassium sparing diuretics depends on higher doses of potassium supplements.

Animal studies have shown a significantly larger increase in TBK [27] and in myocardial potassium content in response to an acute intravenous potassium load [28] in potassium depleted animals as compared to normokalemic controls. It should be noted that in these studies TBK also increased in response to an acute potassium load in normokalemic animals.

The present study is the first to demonstrate that TBK can be increased in normokalemic patients who are not treated with non-potassium sparing diuretics by administering lower doses of potassium supplements (average 36 mmol daily), when used in combination with dietary guidance and concomitant inhibition of the renal potassium loss by administration of MRAs. In addition, it is the first study to demonstrate the timing of changes in p-K and TBK.

We observed a numerical increase in p-K of 0.16 mmol/l at the six months follow-up. This is consistent with increases in p-K of 0.16 to 0.30 mmol/l in patients with heart failure treated with MRA and with p-K increases of 0.09–0.19 mmol/l in patients with hypertension treated with potassium supplements [19].

## Clinical perspective

This study suggests a build-up of intracellular potassium levels over months that did not reflect changes in extracellular potassium in a simple manner, as the significant increase in p-K over weeks was attenuated after six months. If the level of potassium in the intracellular space affects the risk of malignant cardiac arrythmias, a greater effect of potassium-increasing treatment should be expected after months rather than weeks. This will be tested in the POTCAST trial.

## Unanswered questions and future research

The participants in the current study were normokalemic at baseline. Thus, we cannot reasonably infer that raising intracellular potassium takes the same amount of time when deposits are depleted. Potter et al. demonstrated a difference in TBK after only a month of potassium supplementation when added to non-potassium sparing diuretics in elderly patients with heart failure [23]. The present trial does not show if the increased TBK is stable or increase further beyond six months.

## Strengths and weaknesses of the study

The current study represents the first randomized trial to measure the impact of targeting high-normal extracellular potassium levels on TBK. The study was open labelled, i.e., the patients were not blinded to treatment regimen, or the hypothesis of the main trial. The TBK and p-K measurements were all, however, analyzed by technicians blinded to study group assignments. Due to the low sample size and heterogeneous study population, larger studies with a number of subgroups are needed to confirm the generalizability of the findings.

## Conclusion

In normokalemic patients dietary guidance combined with potassium supplementation and MRA treatment increased p-K levels within weeks whereas total body potassium increased gradually, and a significant increase was seen six months after commencement of the intervention. The differentially regulated p-K and TBK and the slower increase in TBK reflect the challenge in monitoring the potassium homeostasis and the time it may take until all effects of a potassium-regulating treatment can be anticipated.

## Supporting information

**S1 File. Supplementary appendix.**
(PDF)

**S2 File. Consort 2010 checklist.**
(PDF)

**S3 File. Study protocol.**
(PDF)

**S4 File. POTCAST study protocol.**
(PDF)

## Author Contributions

**Conceptualization:** Ulrik Winsløw, Peter Oturai, Holger Jan Jensen, Niels Risum, Henning Bundgaard, Christian Jøns.

**Data curation:** Ulrik Winsløw, Tharsika Sakthivel, Chaoqun Zheng, Helle Bosselmann, Ketil Haugan, Niels Bruun, Charlotte Larroudé, Kasper Iversen, Hillah Saffi, Emil Frandsen, Peter Oturai, Holger Jan Jensen, Michael Vinther, Niels Risum, Henning Bundgaard, Christian Jøns.

**Formal analysis:** Ulrik Winsløw, Peter Oturai, Niels Risum, Henning Bundgaard, Christian Jøns.

**Funding acquisition:** Niels Risum, Henning Bundgaard, Christian Jøns.

**Investigation:** Ulrik Winsløw, Helle Bosselmann, Niels Bruun, Hillah Saffi, Peter Oturai, Holger Jan Jensen, Niels Risum, Henning Bundgaard, Christian Jøns.

**Methodology:** Ulrik Winsløw, Ketil Haugan, Kasper Iversen, Peter Oturai, Holger Jan Jensen, Niels Risum, Henning Bundgaard, Christian Jøns.

**Project administration:** Tharsika Sakthivel, Chaoqun Zheng, Helle Bosselmann, Ketil Haugan, Niels Bruun, Charlotte Larroudé, Kasper Iversen, Emil Frandsen, Peter Oturai, Holger Jan Jensen, Michael Vinther, Niels Risum, Henning Bundgaard, Christian Jøns.

**Software:** Holger Jan Jensen.

**Supervision:** Kasper Iversen, Peter Oturai, Holger Jan Jensen, Niels Risum, Henning Bundgaard, Christian Jøns.

**Validation:** Ulrik Winsløw.

**Visualization:** Ulrik Winsløw.

**Writing – original draft:** Ulrik Winsløw.

**Writing – review & editing:** Tharsika Sakthivel, Chaoqun Zheng, Helle Bosselmann, Ketil Haugan, Niels Bruun, Charlotte Larroudé, Kasper Iversen, Hillah Saffi, Emil Frandsen, Peter Oturai, Holger Jan Jensen, Michael Vinther, Niels Risum, Henning Bundgaard, Christian Jøns.

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
