## [Decision Letter · Decision Letter 0]

6 Mar 2023

PONE-D-23-02299Treatment-induced increase in total body potassium in patients at high risk of ventricular arrhythmias; a randomized POTCAST substudyPLOS ONE

Dear Dr. Winsløw,

Thank you for submitting your manuscript to PLOS ONE. After careful consideration, we feel that it has merit but does not fully meet PLOS ONE’s publication criteria as it currently stands. Therefore, we invite you to submit a revised version of the manuscript that addresses the points raised during the review process.

We look forward to receiving your revised manuscript.

Kind regards,

Yoshihiro Fukumoto

Academic Editor

PLOS ONE

Journal Requirements:

"This study was supported by The Danish Council for Independent Research, The Hartmann Foundation, The Danish Heart Foundation, Snedkermester Sophus Jacobsen og hustru Astrid Jacobsens Fond, and the Novo Nordisk Foundation. The funders had no involvement in planning the study design, execution, analysis or interpretation of data.  "

"This study was supported by:

The Danish Council for Independent Research

Grant recipient: CJ

Grant Number: 8020-00399B

Url: https://dff.dk/en

The Hartmann Foundation

Grant recipient: HB

Grant Number: 2019

Url: https://www.hartmannfonden.dk/english/

The Danish Heart Foundation

Grant recipient: HB

Grant Number: 2019

Url: https://hjerteforeningen.dk/

Snedkermester Sophus Jacobsen og hustru Astrid Jacobsens Fond

Grant recipient: HB

Grant Number: 2019

Url: https://sophusjacobsenfond.dk/

The Novo Nordisk Foundation

Grant recipient: NR

Grant Number: NNF20OC0064048

Url: https://novonordiskfonden.dk/

"NO authors have competing interests"

4.We note that you have indicated that data from this study are available upon request. PLOS only allows data to be available upon request if there are legal or ethical restrictions on sharing data publicly. For information on unacceptable data access restrictions, please see http://journals.plos.org/plosone/s/data-availability#loc-unacceptable-data-access-restrictions. 

Reviewers' comments:

Reviewer's Responses to Questions

**Comments to the Author**

1. Is the manuscript technically sound, and do the data support the conclusions?

Reviewer #1: Partly

Reviewer #2: No

2. Has the statistical analysis been performed appropriately and rigorously? 

Reviewer #1: No

Reviewer #2: No

3. Have the authors made all data underlying the findings in their manuscript fully available?

Reviewer #1: No

Reviewer #2: No

4. Is the manuscript presented in an intelligible fashion and written in standard English?

Reviewer #1: Yes

Reviewer #2: Yes

5. Review Comments to the Author

Reviewer #1: The manuscript addresses a potentially interesting topic. The study is single-center and this may limit its usefulness. Similarly, the methods are rather basic and not fully discussed. Some comments follow.

1. Please, upload the code used to get the results. This ensures the reproducibility of the results and allows the reviewer to check for the correctness of the results.

2. The sample size is rather small. This may strongly affect statistical inference. Even though basic tests are employed, no matter which parametric approach is considered, the assumptions underlying the statistical methods must be carefully checked and discussed.

3. I appreciate the use of mixed effects modelling as the data have a clear longitudinal structure. Nevertheless, the results are not well discussed. Firstly, it is rather unclear what the outcome is. The paper discusses mean differences at multiple times, and the regression modelling is swept under the carpet. Again, model's assumptions must be checked, investigated and discussed. A residual analysis must be provided and results must be shown to support your statements. Moreover, I am wondering how the Gaussian assumption for the random effects is tenable for such a small sample. Overall, it is rather unclear if confounders are considered or not, and the empircal specification of the linear predictor is not discussed.

Reviewer #2: This is an interesting study considering that serum potassium level is significantly associated with the cardiovascular event risk. Another important point suggested from the study is that p-K level and TBK are rather differently regulated and do not behave parallelly in the living body homeostasis.

However, the reviewer has concerns as below.

#1. Only value differences were plotted in Figure 2. The measurement values should definitely be separately presented in each group with statistical standard deviations.

#2.The authors’ idea about the sample size calculation is questionable. The reviewer agree with the idea that the study was powered to detect an increase in TBK of 5%. However, as shown in Supplemental Figure 2, the mean difference between measurements in the same sample was 0.024 g/kg body weight, which may up-to 1.6% of the absolute value. Considering the measurement error, the reviewer can not believe that the sample size calculation is sufficient to interpret the results. The sample size calculation should definitely be revised. I guess more number of samples should be included to lead a conclusion.

6. PLOS authors have the option to publish the peer review history of their article (what does this mean?). If published, this will include your full peer review and any attached files.

Reviewer #1: No

Reviewer #2: No

---

## [Author Response · Author response to Decision Letter 0]

9 Jun 2023

Reviewer #1: The manuscript addresses a potentially interesting topic. The study is single-center and this may limit its usefulness. Similarly, the methods are rather basic and not fully discussed. Some comments follow.

1. Please, upload the code used to get the results. This ensures the reproducibility of the results and allows the reviewer to check for the correctness of the results.

Response: Certainly. The code for main results is presented below.

Using the Statistics software R version 4.1.0 and the open-source package lme4:

#loading package for mixed linear regression analysis

library(lme4) # "golden standard" for mixed-effects modelling in R 

#Creating a dataframe with long format data required for mixed regression analysis.

glm_tbk<-c(tbk_baseline,tbk_6weeks, tbk_6months)

glm_pk<-c(pk_baseline, pk_6weeks,pk_6months)

glm_fu<-gl(3,14)

glm_fu<-as.numeric(glm_fu)

glm_pt_nr<-c(1:14,1:14,1:14)

glm_group<-c(group, group, group)

glm_group<-as.factor(glm_group)

glm_koen<-c(koen, koen, koen)

glm.data<-data.frame(glm_tbk, glm_pk, glm_fu, glm_pt_nr, glm_group)

#changing the unit of glm_tbk from g/kg to mg/kg 

glm.data$glm_tbk<-glm.data$glm_tbk*1000

#changing class of glm_fu and glm_group to factor

glm.data$glm_fu<-as.factor(glm.data$glm_fu)

glm.data$glm_group<-as.factor(glm.data$glm_group)

#linear mixed regressio nmodel with patient ID (glm_pt_nr) as random effect.

lmer<-lmer(glm_tbk~glm_fu*glm_group+(1|glm_pt_nr), data=glm.data3, REML=F)

#visual inspection of residual normality

qqnorm(resid(lmer))

qqline(resid(lmer))

#formal statistical test for residual normality

shapiro.test(resid(lmer))

#extracting model results

summary(lmer)

confint(lmer)

coef(lmer)

#linear mixed regression model of p-K with patient ID (glm_pt_nr) as random effect.

lmer2<-lmer(glm_pk~glm_fu*glm_group+glm_group+(1|glm_pt_nr), data=glm.data3, REML=F)

#visual inspection of residual normality

qqnorm(resid(lmer2))

qqline(resid(lmer2))

#formal statistical test for residual normality

shapiro.test(resid(lmer2))

#extracting model results

summary(lmer2)

confint(lmer2)

coef(lmer2)

2. The sample size is rather small. This may strongly affect statistical inference. Even though basic tests are employed, no matter which parametric approach is considered, the assumptions underlying the statistical methods must be carefully checked and discussed.

3. I appreciate the use of mixed effects modelling as the data have a clear longitudinal structure. Nevertheless, the results are not well discussed. Firstly, it is rather unclear what the outcome is. The paper discusses mean differences at multiple times, and the regression modelling is swept under the carpet. Again, model's assumptions must be checked, investigated and discussed. A residual analysis must be provided and results must be shown to support your statements. Moreover, I am wondering how the Gaussian assumption for the random effects is tenable for such a small sample. Overall, it is rather unclear if confounders are considered or not, and the empirical specification of the linear predictor is not discussed.

Response: Thank you for these two important comments on sample size and statistical modelling. We agree that checking the assumptions behind any statistical test is of vital importance. The underlying test for the main results in the present paper is the linear mixed random effects regression model and the assumptions was tested during data analysis. The code is shown in comment 1. We have included a residual analysis in the manuscript:

Page 7, line 167: The assumption of normal distribution of model residuals was tested by Shapiro-Wilk test. For changes in p-K the W statistic was 0.97, P=0.39 and for TBK the W statistic was 0.96, P=0.11. 

The outcome was predefined as between-group difference in changes in p-K and TBK from baseline to six weeks and from baseline to six months. This has been made clearer on page 6 line 134: Main outcome measures: Between-group difference in changes in TBK and p-K from baseline to six weeks, and from baseline to six months.

As for confounding factors and generalizability, the study is a randomized trial and such trial of its kind. However, we do agree that the relatively low sample size, the between-study heterogeneity, may affect the generalizability of the results. This has now been addressed in the limitations section on page 14 line 276: 

Changes made to the manuscript: 

Due to the low sample size and heterogeneous study population, larger studies with a number of subgroups are needed to confirm the generalizability of the findings. 

Reviewer #2: This is an interesting study considering that serum potassium level is significantly associated with the cardiovascular event risk. Another important point suggested from the study is that p-K level and TBK are rather differently regulated and do not behave parallelly in the living body homeostasis.

However, the reviewer has concerns as below.

#1. Only value differences were plotted in Figure 2. The measurement values should definitely be separately presented in each group with statistical standard deviations.

Response: Thank you for this comment. We chose to keep the study simple by only presenting differences in outcome between the two groups in the main paper. We realize that some readers may be interested to know the changes in each group separately along with standard deviations and this data has been added to a table in the supplementary appendix (Supplementary table 1). We have added a sentence to the manuscript in order to refer readers to the table on page 11 line 207.

Changes made to the manuscript:

The mean TBK and p-K in each study group at baseline, six weeks, and six months are shown in S1 Table. 

#2.The authors’ idea about the sample size calculation is questionable. The reviewer agree with the idea that the study was powered to detect an increase in TBK of 5%. However, as shown in Supplemental Figure 2, the mean difference between measurements in the same sample was 0.024 g/kg body weight, which may up-to 1.6% of the absolute value. Considering the measurement error, the reviewer can not believe that the sample size calculation is sufficient to interpret the results. The sample size calculation should definitely be revised. I guess more number of samples should be included to lead a conclusion.

Response: Thank you for this comment. The sample size calculation was performed prior to the initiation of the study and thus, naturally holds estimations of the study outcome. As no prior trial has been made on the same study population some assumptions had to be made. Thus, we assumed that …… and standard deviations for the sample size calculation were achieved from …. 

We all prefer large studies – but we also have to acknowledge the ethics - that it is considered un-ethical to expose more patients to trial examinations than what has been shown to be necessary to reach a conclusion in the specific study. The observed changes came quite close to the estimated changes. TBK increased slightly more than estimated (0.083 g/kg) while the standard deviation of the paired differences was 0.05. A future study with a sample size calculation based on these numbers would need 6 patients in each group.

---

## [Decision Letter · Decision Letter 1]

27 Jun 2023

PONE-D-23-02299R1Treatment-induced increase in total body potassium in patients at high risk of ventricular arrhythmias; a randomized POTCAST substudyPLOS ONE

Dear Dr. Winsløw,

Thank you for submitting your manuscript to PLOS ONE. After careful consideration, we feel that it has merit but does not fully meet PLOS ONE’s publication criteria as it currently stands. Therefore, we invite you to submit a revised version of the manuscript that addresses the points raised during the review process.

We look forward to receiving your revised manuscript.

Kind regards,

Yoshihiro Fukumoto

Academic Editor

PLOS ONE

Journal Requirements:

Additional Editor Comments:

This academic editor considers that the authors well performed the present study, and that they well responded to the reviewers’ comments. This editor has an additional comment as described below.

Major comment:

1. The authors should add why they performed this study in this sample size in the “Ethics”, as responded in comment #2 of Reviewer #2 (that it is considered un-ethical to expose more patients to trial examinations than what has been shown to be necessary to reach a conclusion in the specific study).

Reviewers' comments:

Reviewer's Responses to Questions

**Comments to the Author**

1. If the authors have adequately addressed your comments raised in a previous round of review and you feel that this manuscript is now acceptable for publication, you may indicate that here to bypass the “Comments to the Author” section, enter your conflict of interest statement in the “Confidential to Editor” section, and submit your "Accept" recommendation.

Reviewer #1: All comments have been addressed

Reviewer #2: (No Response)

2. Is the manuscript technically sound, and do the data support the conclusions?

Reviewer #1: (No Response)

Reviewer #2: No

3. Has the statistical analysis been performed appropriately and rigorously? 

Reviewer #1: (No Response)

Reviewer #2: Yes

4. Have the authors made all data underlying the findings in their manuscript fully available?

Reviewer #1: (No Response)

Reviewer #2: No

5. Is the manuscript presented in an intelligible fashion and written in standard English?

Reviewer #1: (No Response)

Reviewer #2: Yes

6. Review Comments to the Author

Reviewer #1: (No Response)

Reviewer #2: Many of the reviewers' comments and requests have not been addressed. The revision made by the authors is not sufficient to overcome the fundamental problems of the study.

7. PLOS authors have the option to publish the peer review history of their article (what does this mean?). If published, this will include your full peer review and any attached files.

Reviewer #1: No

Reviewer #2: No

---

## [Author Response · Author response to Decision Letter 1]

30 Jun 2023

In this revision we have responded to the editor's advice to add a comment on the ethics behind the chosen sample size. We have added the following to the manuscript's ethics section (page 7 line 160):

“The limited size of the study was solely based on the sample size calculation as it is, due to ethical considerations, recommended to avoid exposing more patients to trial examinations than what has been shown to be necessary to reach a conclusion.“

---

## [Editor Report · Decision Letter 2]

5 Jul 2023

Treatment-induced increase in total body potassium in patients at high risk of ventricular arrhythmias; a randomized POTCAST substudy

PONE-D-23-02299R2

Dear Dr. Winsløw,

We’re pleased to inform you that your manuscript has been judged scientifically suitable for publication and will be formally accepted for publication once it meets all outstanding technical requirements.

Kind regards,

Yoshihiro Fukumoto

Academic Editor

PLOS ONE
---

## [Editor Report · Acceptance letter]

10 Jul 2023

PONE-D-23-02299R2 

Treatment-induced increase in total body potassium in patients at high risk of ventricular arrhythmias; a randomized POTCAST substudy 

Dear Dr. Winsløw:

I'm pleased to inform you that your manuscript has been deemed suitable for publication in PLOS ONE. Congratulations! Your manuscript is now with our production department. 

Kind regards, 

on behalf of

Dr. Yoshihiro Fukumoto 

Academic Editor

PLOS ONE